# Characterization of Matrix Metalloprotease-9 Gene from Nile tilapia (*Oreochromis niloticus*) and Its High-Level Expression Induced by the *Streptococcus agalactiae* Challenge

**DOI:** 10.3390/biom10010076

**Published:** 2020-01-03

**Authors:** Fu-Rui Liang, Qin-Qing Wang, Yun-Lin Jiang, Bei-Ying Yue, Qian-Zhi Zhou, Jiang-Hai Wang

**Affiliations:** 1Guangdong Provincial Key Laboratory of Marine Resources and Coastal Engineering, School of Marine Sciences, Sun Yat-Sen University, Guangzhou 510006, China; liangfurui@yeah.net (F.-R.L.); jiangyl7@mail2.sysu.edu.cn (Y.-L.J.); yueby3@mail2.sysu.edu.cn (B.-Y.Y.); zhouqzhi@mail2.sysu.edu.cn (Q.-Z.Z.); 2Southern Marine Science and Engineering Guangdong Laboratory (Zhuhai), Zhuhai 519000, China; 3South China Sea Bioresource Exploitation and Utilization Collaborative Innovation Center, School of Marine Sciences, Sun Yat-Sen University, Guangzhou 510006, China

**Keywords:** matrix metalloproteinase-9, Nile tilapia, *Streptococcus agalactiae*, heterologous expression, qPCR

## Abstract

The bacterial diseases of tilapia caused by *Streptococcus agalactiae* have resulted in the high mortality and huge economic loss in the tilapia industry. Matrix metalloproteinase-9 (MMP-9) may play an important role in fighting infection. However, the role of MMP-9 in Nile tilapia against *S. agalactiae* is still unclear. In this work, MMP-9 cDNA of Nile tilapia (*NtMMP-9*) has been cloned and characterized. *NtMMP-9* has 2043 bp and encodes a putative protein of 680 amino acids. NtMMP-9 contains the conserved domains interacting with decorin and inhibitors via binding forces compared to those in other teleosts. Quantitative real-time-polymerase chain reaction (qPCR) analysis reveals that NtMMP-9 distinctly upregulated following *S. agalactiae* infection in a tissue- and time-dependent response pattern, and the tissues, including liver, spleen, and intestines, are the major organs against a *S. agalactiae* infection. Besides, the proteolytic activity of NtMMP-9 is also confirmed by heterologous expression and zymography, which proves the active function of NtMMP-9 interacting with other factors. The findings indicate that NtMMP-9 was involved in immune responses against the bacterial challenge at the transcriptional level. Further work will focus on the molecular mechanisms of NtMMP-9 to respond and modulate the signaling pathways in Nile tilapia against *S. agalactiae* invasion and the development of NtMMP-9-related predictive biomarkers or vaccines for preventing bacterial infection in the tilapia industry.

## 1. Introduction

Matrix metalloproteinases (MMPs) are a family of 28 zinc-dependent endopeptidases [1], and they are widely distributed in all kingdoms of life [2]. MMPs are secreted as a form of zymogens, and can be activated through the conformational change or cysteine-switch proteolysis [3], which contributes to cleave an extracellular matrix (ECM) and to regulate pathological processes such as inflammation and innate immune defense [4,5] by degradation of the ECM, alteration of cell–cell and cell–ECM interactions, and cleavage of membrane proteins of cell surface and cleavage of proteins in the extracellular environment. Meanwhile, tissue inhibitors of matrix metalloproteinases (TIMPs) inhibit the MMP activity [6]. As one of the well-known MMP family members, MMP-9 can be activated by proteases, and plays an important role in biological processes. The expression of MMP-9 has been found to upregulate in response to proinflammatory cytokines (e.g., IL-1β) in cells such as leukocytes, macrophages, and monocytes [4]. Moreover, inflammatory leukocytes are reported to increase drastically, which leads to a longer period of inflammation in *MMP-9* knockout mice following an allergen challenge [7]. These indicate that MMP-9 has significant effects on inflammatory processes.

Many investigators have paid an increasing attention to the immune responses of MMP-9. It is shown that innate immune cells and ECM hydrolysis as triggering steps are put into the classical paradigms of autoimmunity [8]. Distinctly higher ratios of MMP-9/MMP-2 and activated MMP-9/proMMP-9 are found in the sera of achalasia patients than the controls, and MMP-9 can be used as an innate immune effector for binding to novel substrates in achalasia [9]. In addition, the immunoreactivity of MMP-9 improves to the point of significance in stratum radiatum and the molecule layer of rat hippocampus during the acute stress response [10]. Recently, many reports on the MMP-9 function in fishes has also been investigated, and the overexpression of MMP-9 is determined to protect fishes, such as yellow catfish (*Pelteobagrus fulvidraco*) [4], zebrafish (*Danio rerio*) [11,12], and grass carp (*Ctenopharyngodon idella*) [13] from the infection of pathogens. Thus, MMP-9 seems to be involved in immune responses against bacterial pathogens. To date, the *MMP-9* gene has been cloned in various fishes, including yellow catfish (*P. fulvidraco*) [4], grass carp (*D. rerio*) [13], and channel catfish (*Ictalurus punctatus*) [14].

Nile tilapia (*Oreochromis niloticus*) is the second most farmed fish worldwide due to its rapid growth, high fecundity, and nutritive values [15,16,17]. However, the infectious diseases of Nile tilapia caused by *S. agalactiae* with pathogenic symptoms (like hemorrhage) bring about severe morbidity and mortality, which leads to a huge economic loss for the tilapia industry [18,19]. Some measures have been taken to improve innate immune responses of tilapia against the *S. agalactiae* challenge, e.g., adding oil additives [18] or vaccines (such as GapA, Sip, and FbsA/α-enolase) [17,20,21]. Moreover, a few immune proteins (e.g., CatL and CatB) are found to activate immune signaling pathways based on the information of transcriptome [19] and microRNAs [15]. MMP-9 is likely to be involved in immune responses of tilapia. However, the role of MMP-9 in tilapia against *S*. *agalactiae* infection is still not well understood. Fortunately, the MMP-9 study is more convenient due to the use of the sequenced genome and transcriptome of tilapia [19,22]. In this study, we cloned NtMMP-9 and subsequently analyzed the interactions between NtMMP-9 and TIMP-2 or decorin (DCN). The expression patterns of NtMMP-9 at different time points and in different tissues of Nile tilapia were also checked using qPCR. Moreover, the heterologous expression of NtMMP-9 in *Escherichia coli* was employed to detect the proteolytic activity in vitro. Our results may reveal the response of NtMMP-9 in Nile tilapia against *S*. *agalactiae* infection, and further explore whether NtMMP-9 can be used as an indicator for preventing and curing bacterial diseases of tilapia.

## 2. Materials and Methods

### 2.1. Fish Cultivation and Pathogen-Free Validation

A total of 80 juvenile fish (*O. niloticus*) were purchased from a national fish haven in Guangzhou (China), which were 45 days of age with the weight of 10 ± 0.1 g. The number of fish used in this study was 76. Before the experiments, they were precultured in a fish tank (150 L) with recirculating aerated water at 20 °C for 7 days, and fed with commercial pellets once a day as previously described [16]. Afterwards, three fish were selected randomly and dissected to obtain six tissues (including brain, spleen, gill, head kidney, intestines, and liver) for a pathogen-free detection on Brain Heart Infusion (BHI) plates, respectively [16]. The experimental protocol of fish (*O. niloticus*) was in accordance with the National Institutes of Health guide for the care and use of laboratory animals, and it was approved by the Animal Ethical and Welfare Committee of Sun Yat-Sen University with the approval number IACUC-17-0903 on 15 September 2017.

### 2.2. RNA Extraction and NtMMP-9 Cloning

The fish were euthanized with tricaine methanesulfonate (MS-222, Sigma-Aldrich Co., Saint-Louis, Mo, USA) for 5 min to collect tissues. The tissue samples of spleen for cloning the *NtMMP-9* gene and six tissues mentioned above for further qPCR detection were placed in the sample protector (TaKaRa, Dalian, China) to protect the integrity of total RNA and stored at −80 °C until use. The total RNA was extracted from tissues using the MiniBEST Universal RNA Extraction Kit (TaKaRa, Dalian, China). To clone the *NtMMP-9* gene, the total RNA extracted from the spleen tissue of three fish was used to synthesize cDNA with M-MLV reverse transcriptase using the first-strand cDNA Synthesis Kit (ProbeGene, Jiangsu, China).

The *NtMMP-9* gene was PCR-amplified from cDNA using two pairs of primers (NMP9-F1 and NMP9-R1; NMP9-F2 and NMP9-R2) (Table 1) designed according to the submitted genome (GenBank accession No. GCA_001858045.3, 1005.68 Mb) and MMP-9 mRNA (GenBank accession No. XM_003448139.5, 3166 bp) of *O. niloticus* in NCBI using primer 5. The PCR reaction system contained 1 μL of cDNA (3 μg/μL), 25 μL of 2 × Ultra-Pfu Master Mix (ProbeGene, Jiangsu, China), 1 μL of each primer (10 mM), and RNase-free ddH_2_O added to 50 μL. The amplification process was described as follows: initial denaturation for 10 min at 94 °C, 30 cycles (including denaturation for 30 s at 94 °C, annealing for 30 s at 52 °C, and extension for 1 min at 72 °C) and final extension at 72 °C for 3 min. The PCR product was detected and purified as the described method [16]. The purified NtMMP-9 fragment was then ligated into the pMD-19T vector (TaKaRa, Dalian, China) for sequencing, which was later transformed into *E. coli* DH5α competent cells (TaKaRa, Dalian, China).

### 2.3. Bioinformatic Analysis of NtMMP-9

The cDNA sequence and deduced amino acids (aa) of NtMMP-9 were analyzed using BLAST (https://blast.ncbi.nlm.nih.gov/Blast.cgi) and an Expert Protein Analysis System (http://prosite.expasy.org/). The alignment of multiple amino acid sequences was carried out by Clustal X 2.1. A phylogenetic tree of MMP-9s derived from different species was also constructed by MEGA 6.0 using the neighbor-joining algorithm. The protein–protein interactions of active NtMMP-9 (aNtMMP-9, 114–680 aa) and other proteins were conducted by STRING 10.5 (http://string-db.org/). The molecular docking of aNtMMP-9 and TIMP-2 or DCN was also predicted by the ZDOCK server (http://zdock.umassmed.edu/), and using software PyMOL 17.6 to analyze models.

### 2.4. Challenge Experiment with S. agalactiae

The pathogenic strain *S. agalactiae* CCARM 0211 (SYSU, China) used in this study was deposited in the Sun Yat-Sen University culture collection of microorganisms. It was incubated on BHI plates at 28 °C for 36 h. When the OD_600_ of *S. agalactiae* growing in the liquid BHI medium (28 °C, 220 rpm) reached 1.0, cells were harvested by centrifugation (8000 rpm, 5 min) and washed with a phosphate buffer solution (PBS) (pH 7, 10 mM) 3 times. It was then resuspended with the same PBS for obtaining the cell solution (10^7^ CFU/mL). This concentration (LD_50_) was introduced from the result of intraperitoneal challenge to fish after 7 days. Importantly, all fish were not fed for 3 days before the challenge experiment and during the infection period. Fish were randomly divided into three groups: (1) healthy group; (2) control group processed by intraperitoneal injection with the PBS (10 mM, 200 μL); and (3) challenged group treated with *S. agalactiae* (200 μL) by intraperitoneal injection. Ten fish in each group were guaranteed; the control and challenged groups were performed in triplicate, respectively. To determine the transcriptional level of *NtMMP-9,* six tissues mentioned above were collected from three healthy fish. The fish tissues from the control or challenged group were simultaneously isolated after injection at different time points (4, 24, 48 and 72 h). All obtained tissues were stored at −80 °C with protectors for RNA extraction and further study. All fish used in this study are shown in Appendix A.

### 2.5. qPCR Analysis of NtMMP-9 under the S. agalactiae Challenge

The total RNA extraction was carried out as described above, and cDNA synthesis was performed using the PrimeScript^TM^ RT reagent Kit with gDNA Eraser (Perfect Real Time, TaKaRa, Dalian, China) according to the manufacturer’s instruction. Then, the qPCR assay was tested using the CFX96 Touch™ Real-Time PCR detection system (Bio-Rad, Hercules, CA, USA) with the SYBR^®^ Premix Ex Taq^TM^ II (TaKaRa, Dalian, China) and primers qNMP9-F and qNMP9-R (Table 1). The thermal cycling and melting profiles of qPCR were performed as per the reported method [23]. Both 18S rRNA and *UBCE* (ubiquitin-conjugating enzyme gene) were selected as the reference genes due to their special stability at the expression level across various tissues of Nile tilapia [24]. The relative transcriptional levels of *NtMMP-9* in different tissues of the healthy group or challenged group fish were normalized to the transcriptional levels of *NtMMP-9* in the intestines from healthy fish or the control tissues using the 2^−^^ΔΔCt^ method [23,25], respectively.

### 2.6. Heterologous Expression of NtMMP-9

The target active domain of NtMMP-9 (aNtMMP-9, 114‒680 aa) was synthesized after codon optimization by Generay Biotech Co., Ltd. (Shanghai, China), and ligated into pET-28a (+) vector containing the kanamycin resistance gene KanR (Novagen, Madison, WI, USA) to construct the expression vector pET-28a/*aNtMMP-9* (Appendix A). Then, the target vector was transformed into different competent cells including Rosetta-gami (DE3) (WeiDi Biotechnology Co., Ltd., Shanghai, China), BL21 (DE3) and BL21 (DE3) pLysS (Transgen, Beijing, China) by the calcium chloride (CaCl_2_) heat shock method [26] for heterologous expression, respectively.

Positive transformants were picked out from LB (Luria-Bertani, Beijing Solarbio Science & Technology Co., Ltd., Beijing, China) plates with kanamycin (50 mg/mL), and a target-positive transformant containing pET-28a/*aNtMMP-9* vector was confirmed by PCR amplification using the primers yNMP9-F and yNMP9-R with a 1716-bp fragment (Table 1 and Appendix A) and sequencing. The large-scale cultivation (1 L) was conducted after the target clone grew in the LB broth (10 mL) with kanamycin at 37 °C and 220 rpm overnight. When the OD_600_ reached 0.5, the target one was induced to express aNtMMP-9 by adding IPTG (TakaRa, Dalian, China) and simultaneously switching 37 °C to 16 °C for 12 h. Thereafter, the cells obtained by centrifugation (8000 rpm, 5 min) were resuspended in the lysis buffer (containing 50 mM Tris, 3 mM β-mercaptoethanol, 0.3 M NaCl, and 0.1% Triton-100, pH 7) with 1 × ProteinSafe^TM^ Protease Inhibitor Cocktail (Transgen, Beijing, China), and lysed using a Sonicator S-4000 (Misonix, Farmingdale, NY, USA) [16]. The supernatant was then purified by TALON Superflow (GE Healthcare, Chicago, IL, USA). Nontarget proteins were removed using imidazole buffers (5 mL) at different concentrations (20, 30, and 40 mM) respectively, while the target one was eluted using the imidazole buffer (200 mM, 0.5 mL). The purified aNtMMP-9 was ultimately frozen by liquid nitrogen and stored at −80 °C. SDS-PAGE (sodium dodecyl sulfate-polyacrylamide gel electrophoresis) determination and Western Blotting assay were also carried out as previously described [27].

### 2.7. Protease Activity Assay

The protease activity of the purified aNtMMP-9 was tested by gelatin zymography according to the described method [3]. Band intensity was also monitored using the software ImageJ 1.48 (National Institutes of Health, Bethesda, MD, USA) and the formed band intensity represented the protease activity.

### 2.8. Statistical Analysis

The statistical analysis was performed by one-way analysis of variance (ANOVA) and the Dunnett’s post-test using SPSS 22.0 (SPSS Inc., Chicago, IL, USA) [16]. All results were presented as the mean ± SD. The significant difference was represented by asterisk (* 0.01 < *p* < 0.05; and ** *p* < 0.01).

## 3. Results and Discussion

### 3.1. Cloning and Characterization of NtMMP-9

All these tissues were firstly found to be no pathogens on BHI plates (data not shown). Then, all the experiments were accordingly conducted. The *NtMMP-9* gene was cloned according to the information of genome and MMP-9 mRNA of *O. niloticus* as described in the methods section. The gene and encoded amino acids were predicted by using BLAST and an Expert Protein Analysis System. The results showed that MMP-9 has 2043 bp and encodes a putative protein containing 680 amino acid residues (aa) with the predicted molecular mass of 76 kDa (Figure 1). The deduced NtMMP-9 had the typical structural characteristics of MMP family members, including the N-terminal signal peptide (1–21 aa), propeptide domain (38–96 aa), ZnMc domain (114–446 aa), and hemopexin-like repeats (HX domain) (494–674 aa), as well as a low complexity region (450–485 aa). The highly conserved sequence PRCGVPD was found in NtMMP-9 (Figure 1), which agrees with the PRCXXPD motif (where X represents any aa) in MMPs [1]. This motif is regarded as a cysteine switch at the terminal of propeptide, and the cysteine switch is capable of coordinating the interaction between zinc atoms and cysteine thiolate via relieving chelation to regulate the zymogen form of MMPs [28].

The signal peptide is lowercased, and the propeptide is boxed with black. FN2 domain is underlined by double arrows, and the Zn^2+^-binding motif is labeled with ellipse. HX domain is also marked in bold, and the low complexity region is shaded and boxed.

The ZnMc domain consists of a conserved motif (HEFGHALGLDH, 402–412 aa) and three repeats of fibronectin type-II (FN2) domain (Figure 1). This motif is the catalytic domain of MMP-9s, similar to the conserved zinc-binding motif (HEXXHXXGXXH, where X represents any aa) of MMPs [1,2]. The zinc-binding motif chelates the active site of Zn^2+^ through three histidine residues [2]. This may be an important cornerstone for performing the immune function of NtMMP-9. FN2 domain (locating 225−273 aa, 283−331 aa, and 341−389 aa) is responsible for binding and assisting gelatinases to catalyze substrates, which is only reported in MMP-2 and MMP-9 [29]. Furthermore, six of seven disulfide interdomain bridges are formed among the cysteine residues in the FN2 domain fix (the folded structure) and facilitate pro-MMP-9 secretion [30]. The HX domain (locating 494−538 aa, 540−581 aa, 586−632 aa, and 634−674 aa) near the C-terminal is critical to bind TIMPs or other receptors (e.g., α4β1) [31,32]. The sequence of NtMMP-9 (No. MG766449) was submitted to the NCBI database.

### 3.2. Alignment and Phylogenetic Analysis

MMP-9 has been found to enhance the recruitment of macrophages and plays crucial roles in immune responses of several species, including zebrafish [11,12], carp [33], and mice [34]. The multiple sequence alignment of amino acids of NtMMP-9 and those from different species is noted using Clustal X 2.1. It is clear that NtMMP-9 shows more than 81% identity to MMP-9s of *Siniperca chuatsi* and *Fundulus heteroclitus* and 73−75% identity to those from other species (Appendix A). Primary domains of NtMMP-9, including FN2, ZnMc, and HX domains, were also revealed, implying that MMP-9 is relatively conserved among tilapia and other species.

The molecular evolution of NtMMP-9 among MMP-9s derived from other teleosts was evaluated by constructing a phylogenetic tree using MEGA 6.0. As shown in Figure 2, NtMMP-9 was clustered more closely to the MMP-9 from *Maylandia zebra* among teleosts [4], suggesting that the genetic relationship between *O. niloticus* and *M. zebra* may evolve at the same level. Also, it was revealed that NtMMP-9 was one of MMP-9 family members.

### 3.3. Protein–Protein Interactions and Molecular Docking

The prediction of protein–protein interactions was performed by STRING 10.5. NtMMP-9 interacted with at least 20 inflammatory or immune-related proteins (such as proheparin-binding EGF-like growth factor isoform X1 (HB-EGF), TIMP-2, and DCN) are presented in Figure 3 and Appendix A. HB-EGF can be cleaved and then triggers shedding from tissues by MMP-9, accordingly forming a positive feedback loop to enhance the availability of HB-EGF [35]. The DCN activated by NtMMP-9 modulates inflammation by sequestering growth factors [36]. TIMP-2 binds to the HX domain of NtMMP-9 and reduces its activity. The predicted results indicate that NtMMP-9 may be involved in regulating the activities of various proteins in signaling pathways responding to the *S. agalactiae* challenge.

The molecular docking of aNtMMP-9 with DCN or TIMP-2 was conducted by the ZDOCK server and software PyMOL 17.6; the results are illustrated in Figure 4. It is clear that a compact complex is formed between aNtMMP-9 and DCN via significant binding forces, including hydrophobic interaction, hydrogen bonds, and several noncovalent bonds. The amino acid residues Phe-538, Trp-540, Val-551, Gly-552, and Tyr-553 of aNtMMP-9, are predicted to form strong hydrophobic interactions with residues Leu-327, Phe-328, Val-332, Pro-333, and Leu-353 of DCN (Figure 4A and Appendix A). Moreover, significant hydrogen bonds appear between aNtMMP-9 residues (Lys-377, Tyr-553, and Val-551) and some residues (Glu-336, Leu-327, and Leu-353) of DCN. Besides, three interactions—cation–π, anion–π, and CH–π—also occur between aNtMMP-9 and DCN (Appendix A). The above interactions may lead to the formation of the stable complex between aNtMMP-9 and DCN.

Compared with interactions between aNtMMP-9 and DCN, aNtMMP-9 interacting with TIMP-2 also produces a complex through binding forces, including hydrophobic interactions and hydrogen bonds instead of anion-π (Figure 4B and Appendix A). TIMPs inhibit active MMPs by binding to catalytic sites of MMPs to maintain the balance of MMPs and TIMPs, which benefits the degrading extent of ECM [4,31]. Probably, more hydrogen bonds have a potential to favor the inhibition of TIMP-2 to MMP-9.

Protein–protein interactions reveal that NtMMP-9 interacts with other proteins via binding forces, including hydrophobic interaction (cation–π, CH–π, and anion–π) and hydrogen bonds. Clearly, multiple interactions exhibit between NtMMP-9 and DCN in this study, which may induce proinflammatory signaling, and thereby link with innate immunity and inflammation [37]. The proteolytic activities of MMPs including MMP-9 are often mediated by TIMPs. TIMP-2 combines directly with the HX domain as an inhibitor [30], which probably occurs by the described interactions in Appendix A and Figure 4B. These results provide a strong support on the structural and functional information of NtMMP-9 in participating in the proinflammatory signaling pathway and immune response processes by regulating function-related factors directly or indirectly through various interactions.

### 3.4. Transcriptional Expression of NtMMP-9

It has been confirmed that MMP-9 plays an active role in the initial phase of inflammation and in the later phase of tissue remodeling during the processes of innate immune responses according to the transcriptional expression of MMP-9 from carp [33]. The qPCR detection of NtMMP-9 for all examined tissues of healthy fish using the CFX96 Touch™ Real-Time PCR detection system (Bio-Rad, USA) indicates that the higher transcriptional levels of NtMMP-9 occurred in gill (49.87 ± 0.1 fold), head kidney (27.09 ± 5.86 fold), and spleen (17.34 ± 3.24 fold) compared to that in the intestines (as the control) (Appendix A); whereas they were lower, but still significant, in both the liver and brain. These findings suggest that the transcriptional expression of NtMMP-9 is tissue-dependent, and the highest response occurs in the gill.

To detect the effect of NtMMP-9 on the immune response of tilapia against the *S. agalactiae* challenge, the transcriptional levels of NtMMP-9 in different tissues and different time points were also conducted. In the early response phase (0−4 h), the transcriptional level of NtMMP-9 was significantly upregulated in four tissues, especially the liver, spleen, and brain; a slight decrease appeared in the head kidney (Figure 5). Subsequently, the expression of NtMMP-9 increased and reached the peaks in the liver (90.12 ± 18.23 fold) and brain (84.72 ± 13.63 fold) at 24 h after the *S. agalactiae* challenge. At 48 h after the challenge, the peaks of NtMMP-9 expression also occurred in the intestines (47.75 ± 11.65 fold), spleen (44.26 ± 13.2 fold), and head kidney (2.21 ± 0.78 fold). Interestingly, the NtMMP-9 expression in the gill distinctly enhanced at 4 h after challenge, and reached another peak (6.01 ± 2.1 fold) at 72 h (Figure 5). The data suggest that the NtMMP-9 expression is a tissue- and time-dependent response pattern in Nile tilapia against the *S. agalactiae* challenge. Five tissues, including the intestines, brain, liver, spleen, and gill, are more susceptive to the *S. agalactiae* challenge, and the later three tissues may be the main organs responding to pathogenic challenge. Besides, the intestines of Nile tilapia may recover from the *S. agalactiae* challenge when the NtMMP-9 expression changes from the high to the normal levels.

The tissue-specific expression of NtMMP-9 in healthy fish shows that the highest expression occurred in the gill (Appendix A), which is different from previous studies. For example, MMP-9s exhibited the maximal expression in the head kidney of normal yellow catfish [4] and in the blood of grass carp [13] instead of the gill. Thus, the NtMMP-9 expression seems to be tissue-dependent. Moreover, its responding expression levels varied differently after the *S. agalactiae* challenge for 72 h, exhibiting a time-dependent response pattern. Therefore, NtMMP-9 may involve in the immune system of tilapia against the *S. agalactiae* challenge, just as other fish preventing pathogen invasion [4,13].

The liver, brain, and spleen of Nile tilapia proved to be more susceptive to the *S. agalactiae* challenge due to the rapid response of NtMMP-9 (Figure 5), which is consistent with the damage of immune organs (i.e., spleen and head kidney) of tilapia after the *S. agalactiae* challenge [38]. In histopathology, several pathological changes such as encephalitis, lymphocyte necrosis, and atrophied neuronal cells have been found in fish [39]. The histopathological changes of tilapia have been uncovered in our previous studies [16,23], which are also the same responses of Nile tilapia to the *S. agalactiae* challenge because the laboratory fish were identical and treated on the same conditions. The high expression of MMP-9 was observed in spleen (Figure 5), where necrotic lymphocytes and the breakage of cells appeared in spleen after the *S. agalactiae* challenge [23]. This finding coincides with those of other fish [13,38]. The damage to the tilapia brain was identified with macrophages and cavities in cytoplasm at 72 h [16], suggesting that the brain is a susceptible organ; and its pathological changes may be used to reasonably explain the expression pattern of NtMMP-9 against the *S. agalactiae* challenge. The intestines suffered from hyperplasia and proliferation of epithelial cells and disappearance of the goblet structure [16], which responded to NtMMP-9 expression. Further, the intestines might facilitate the homeostasis of tilapia by mediating NtMMP-9 expression [39]. Interestingly, NtMMP-9 expression in the gill appeared to upregulate, and reached another peak at 72 h again (Figure 5). We deduce that MMP-9 expression varies from different tissues and different time points between hosts and pathogens, and the instability of MMP-9 mRNA or inhibitors may change it [4]. In addition, overexpression of MMP-9 reduces cell apoptosis in mice [40]. These results indicate that NtMMP-9 has a positive role in the immunological regulation of Nile tilapia.

### 3.5. Heterologous Expression and the Enzyme Activity of NtMMP-9

The expressing vector pET-28a/*aNtMMP-9* was recombined for determining the enzyme activity of NtMMP-9. As shown in Appendix A, *E. coli* Rosetta-gami (DE3) competent cells are thought to be more responsible for aNtMMP-9 expression as the host, instead of BL21 and BL21 pLysS. The high capability of cells to successfully express eukaryotic proteins might result from trxB/gor mutations and the presence of T7 lysozyme in Rosetta-gami (DE3), contributing to fold eukaryotic proteins with abundant disulfide bonds, and thus can exist in the cytoplasm as the soluble form [41].

In this study, when aNtMMP-9 expression was induced by adding IPTG (0.5 mM) and the product was purified; clear bands of the target proteins were displayed via SDS-PAGE (Appendix A). Apparent bands formed, revealing that the purified aNtMMP-9 bounded to the anti-His Tag antibody via Western Blotting (Appendix A). The activity levels of MMP-9 were also assessed by zymography [3,42]. A white band of aNtMMP-9 was also found via gelatin zymography with a molecular weight of approximate 63.4 kDa (Appendix A), confirming the protease activity of NtMMP-9 in the biological function.

MMP-9 has an immuno-regulatory function in mice cornea against *Pseudomonas aeruginosa* by proteolysis and upregulating cytokines/chemokines [43]. Furthermore, during the pulmonary infection of mice by *Francisella tularensis*, MMP-9 can generate proinflammatory ECM-derived peptides against the infection [44]. Similarly, *Ciona* MMP-9 is involved in inflammatory responses induced by a lipopolysaccharide based on the elevated mRNA levels and enzymatic activity of MMP-9 [3]. MMP-9 is also the most upregulated gene following the *Listeria* infection in zebrafish, and can be used a protective molecule [12]. Besides, MMP-9 is a strong, independent predictor of atherosclerotic plaque instability [45]. The results suggest that NtMMP-9 (aNtMMP-9) may have a basic function to regulate inflammatory and immune responses of tilapia against a *S. agalactiae* infection.

## 4. Conclusions

The obtained NtMMP-9 in this study has similar conserved domains/motifs to those of MMP-9s in teleosts, and these conserved domains/motifs may bind to different molecules. It is predicted that NtMMP-9 interacts with other factors by several binding forces. During the *S*. *agalactiae* challenge, the upregulation of NtMMP-9 expression was tissue- and time-dependent, at least in checked tissues; the major immune organs are liver, spleen, and intestines. Our data suggest that NtMMP-9 has a basic function to regulate the immune response of tilapia against bacterial infection. Further, NtMMP-9 may be a potential predictive biomarker for promoting the healthy development of tilapia.

## Figures and Tables

**Figure 1 biomolecules-10-00076-f001:**
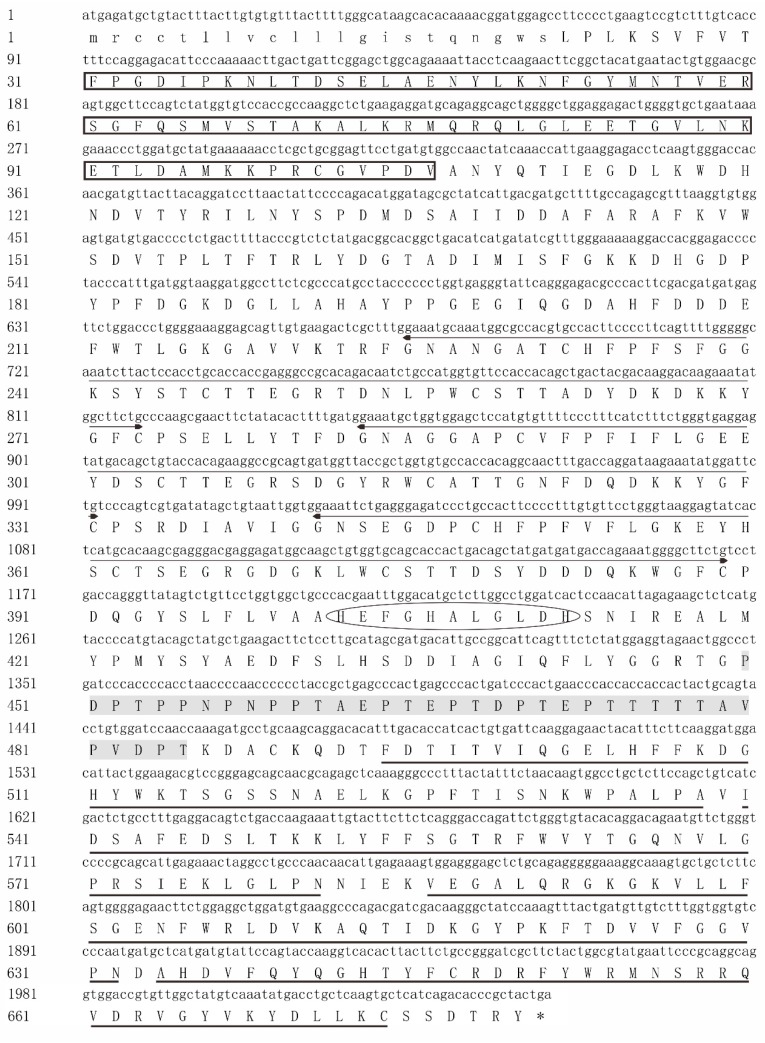
The cDNA sequence and putative amino acids of *NtMMP-9* in Nile tilapia.

**Figure 2 biomolecules-10-00076-f002:**
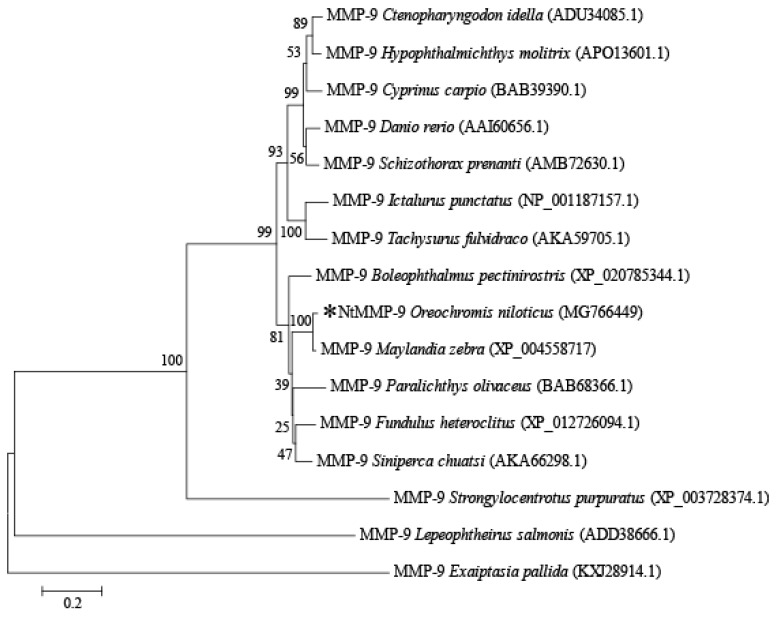
Phylogenetic tree of MMP-9s constructed from different species. Bootstrap values are 1000; and NtMMP-9 is marked with a black star.

**Figure 3 biomolecules-10-00076-f003:**
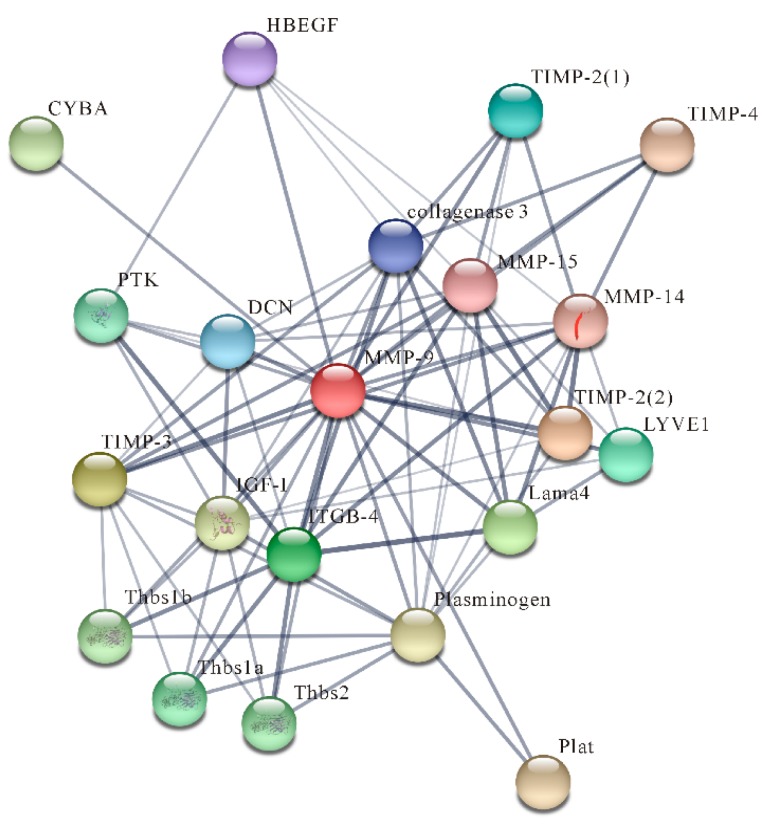
A network of protein–protein interactions created by NtMMP-9 and other proteins.

**Figure 4 biomolecules-10-00076-f004:**
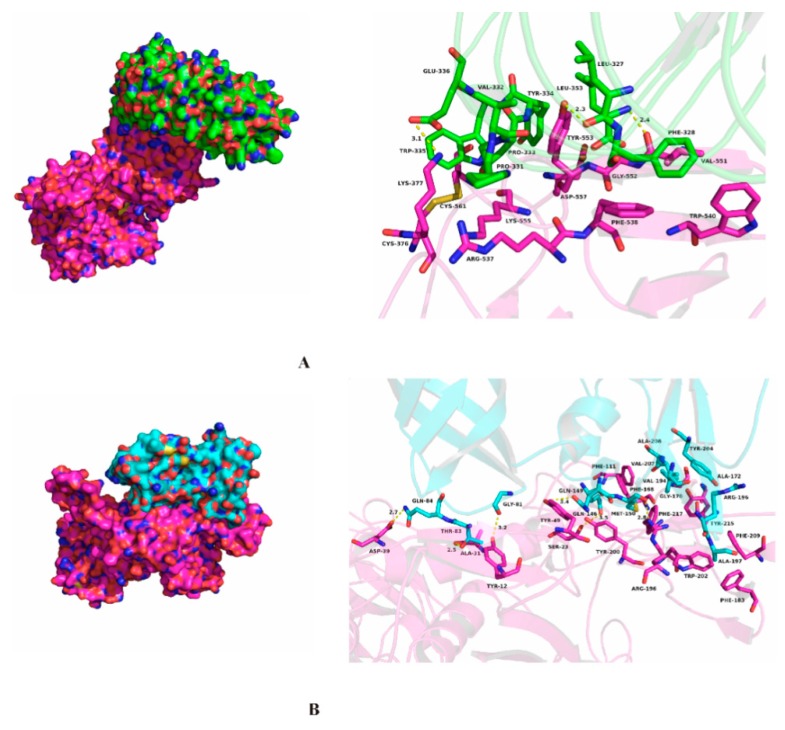
Molecular dockings of aNtMMP-9 and other proteins. (**A**) Molecular docking of aNtMMP-9 (rose red) interacted with DCN (green); and (**B**) molecular docking of aNtMMP-9 (rose red) interacted with TIMP-2 (cyan).

**Figure 5 biomolecules-10-00076-f005:**
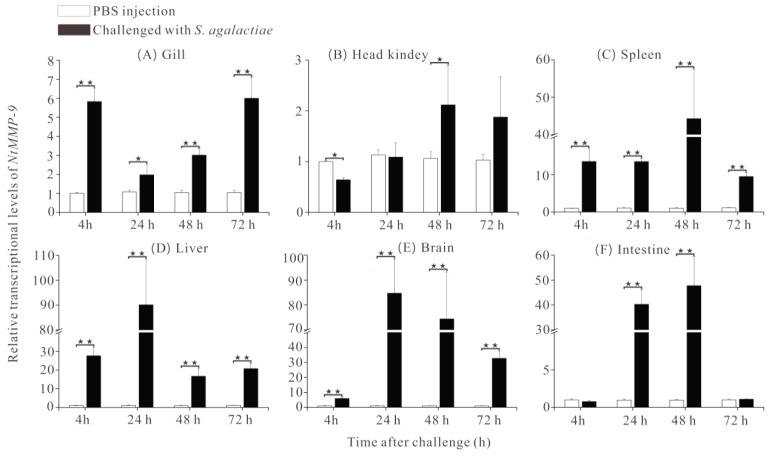
Relative transcriptional levels of NtMMP-9 at different time points in six tissues of Nile tilapia after the *S. agalactiae* challenge. Six tissues are gill (**A**), head kindey (**B**), spleen (**C**), liver (**D**), brain (**E**), intestine (**F**) in sequence. (* 0.01 < *p* < 0.05; and ** *p* < 0.01).

**Table 1 biomolecules-10-00076-t001:** Primers were used for cloning and expressing the *NtMMP-9* gene in this study.

Primer	Sequence (5′–3′)
NMP9-F1	GTGCCGCGCGGCAGCCATATGAGATGCTGTACTTTACTTGTG
NMP9-R1	AAGGAGTATCACTCATGCACAAG
NMP9-F2	GTATCACTCATGCACAAGCGAG
NMP9-R2	CTCAAGTGCTCATCAGACACCCGCTACTGACTCGAGCACCACCA
qNMP9-F	ATGCTTTTGCCAGAGCGTTT
qNMP9-R	TGTCAGCCGTGCCGTCA
yNMP9-F	CATATGGGCGATCTGAAATGGGATC (the underlined bases encode *Nde* I)
yNMP9-R	CTCGAGTTAATAGCGTGTATCGCTTGA (the underlined bases encode *Xho* 1)

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
