# Peer review of "Characterization of Matrix Metalloprotease-9 Gene from Nile tilapia (Oreochromis niloticus) and Its High-Level Expression Induced by the Streptococcus agalactiae Challenge"

_biomolecules, 2020, doi:10.3390/biom10010076_

Round 1
Reviewer 1 Report
This is an important study for the identification and characterization of the Matrix metalloprotease-9 gene from Nile tilapia. The Authors hve done a good study, but there is need for a lot of improvement on the manuscript to be done. it is Clear that the Authors are not native neglish writers and they need help from a native English Write to re-Write the Whole manuscript to make it Clear and easy to understand.
I have highlighted in yellow some of the sentences that require grammatic correction as an example.
MAJOR CONCERNS
Line 77: How many Fish altogether were included in the study?
Line 80: "several fishes", exactly how many? Better to be precise.'
Line 81: What pathogens were screened?
Line 87: What tissue samples were collected here?
Lines 114-132: I suggest that the Authors provide a drawing showing the number of Fish per tank, number of replicates, timeline of sampling and type of samples collected. In the Current state it is confusing to understanding, especially with comparison to lines 77-84.

Author Response
Thank you very much for your good comments. We have made improvement on the manuscript, including the grammar, words, English language, structure, and other places through the whole manuscript.
Line 77: How many fish altogether were included in the study?
Response: Thanks for your good suggestion. In fact, a total of 80 fish (Oreochromis niloticus) were purchased, and 76 fish was used for this study. The more detailed information was given in section Materials and Methods (Lines 85-87) in the revised version and Supplementary Table S1.
Line 80: "several fishes", exactly how many? Better to be precise.'
Response: Thanks for your good suggestion. Here, three fish were randomly selected and dissected to obtain six tissues for pathogen-free detection (Line 89 in the revised version).
Line 81: What pathogens were screened?
Response: All these tissues from three selected-randomly fish were conformed to be no pathogens on BHI plates (data not shown), and then all the experiments were conducted. The BHI medium mainly contains peptone, extract powder of dehydrated calf brain, extract powder of dehydrated buffalo hear, glucose, sodium chloride, and disodium hydrogen phosphate, which provides available carbon and nitrogen sources for Streptococcus spp. and other bacteria like Escherichia coli. As reported in previous studies, the BHI medium has been used to cultivate Streptococcus agalactiae and other bacteria like Aeromonas spp. for the fish Nile Tilapia (Oreochromis niloticus) against S. agalactiae (Meidong et al., 2017; Pasaribu et al., 2018). Pasaribu et al. (2018) have also used the BHI medium to incubate S. agalactiae and to ensure that S. agalactiae is killed and there is no contamination. Therefore, we believe that the BHI medium used in our study to confirm no pathogens in tissues in Nile Tilapia is feasible.
References:
Meidong R, Doolgindachbaporn S, Sakai K, et al. Isolation and selection of lactic acid bacteria from Thai indigenous fermented foods for use as probiotics in tilapia fish Oreochromis niloticus. Aquaculture, Aquarium, Conservation & Legislation, 2017, 10(2): 455-463.
Pasaribu W, Sukenda S, Nuryati S. The efficacy of Nile Tilapia (Oreochromis niloticus) Broodstock and larval immunization against Streptococcus agalactiae and Aeromonas hydrophila. Fishes, 2018, 3(1): 16.
Line 87: What tissue samples were collected here?
Response: The tissue sample of spleen for cloning the NtMMP-9 gene, and six tissues mentioned above (including brain, spleen, gill, head kidney, intestine, and liver) for further qPCR detection were placed in the sample protector (TaKaRa, Japan) to protect the integrity of total RNA and stored at ‒80℃ until use (Lines 97-99 in the revised version).
Lines 114-132: I suggest that the authors provide a drawing showing the number of fish per tank, number of replicates, timeline of sampling and type of samples collected. In the current state,it is confusing to understanding, especially with comparison to lines 77-84.
Response: Thanks for your good comments. We have provided a complete description showing the number of fish per tank, number of replicates, timeline of sampling, and type of samples collected (Supplementary Table S1). The corresponding corrections are also presented in the revised version (Line 85-87, 89, 101, and 135-144).
Reviewer 2 Report
This manuscript describes the characterization of MMP-9 gene in Nile tilapia and the relation of this molecule with the immune response against Streptococcus agalactiae.
The manuscript results could be of interest however the manuscript should be deeply corrected in order to be acceptable for publication. Please find below the major concerns:
English should be deeply edited. It is difficult to follow the manuscript… Incomplete introduction, more background and explanations about MMP-9 as antimicrobial or immunomodulator is needed Results about gene cloning and sequencing are confusing, since only two primer pairs are used for sequencing a gene of approximately 2000 bp, please better explained about it. Also include in the results about the gene and protein predictions the programs that you used for each prediction. Results about transcriptome and molecular docking are not properly explained and their related assays are absent in material and methods section Results about “Heterologous expression and the enzyme activity of NtMMP-9” figure is absent, also the meaning of these results are not clearly explained. I would recommend performing an antibacterial assay. Incomplete and wrongly written figure legends all throughout the text. Statistical tests used should be indicated Conclusions should be further developed.Minor concerns:
Several text miss-writing errors should be corrected (ie changes in letter size and several words underlined throughout the text) Line 95: sequence species origin Line 142: explain what is RATE Line 173: further detail the gene sequencing results, ie: fragments sequence lengths, etc.Author Response
Thank you very much for your good comments. We have made improvement on the manuscript, including the grammar, words, English language, structure, and other places through the whole manuscript.
The manuscript results could be of interest. However, the manuscript should be deeply corrected in order to be acceptable for publication. Please find below the major concerns:
English should be deeply edited. It is difficult to follow the manuscript… Incomplete introduction, more background and explanations about MMP-9 as antimicrobial or immunomodulator is needed.
Response: Thanks for your good suggestion. We have deeply polished our manuscript, including, but not limited to English grammar and incomplete introduction, more background and explanations about MMP-9 as antimicrobial or immunomodulator in section Introduction in the revised reversion (Lines 53-59).
Results about gene cloning and sequencing are confusing, since only two primer pairs are used for sequencing a gene of approximately 2000 bp, please better explained about it.
Response: Thank you for reminding us. We have further described the results on gene cloning and sequencing in the revised version (Lines 106-109 and Table 1).
Also include in the results about the gene and protein predictions the programs that you used for each prediction.
Response: Thanks for your good suggestion. The programs on the gene and protein predictions used in this study have been presented in section Methods in the revised version; and each prediction, including the used program has been added in section Results in the revised version (Lines 106-109, 118-126, 182-184, 195-197, 232, 237, 243, 254-255, and 289-290).
Results about transcriptome and molecular dockingare not properly explained and their related assays are absent in material and methods section.
Response: Thank you for reminding us. We have checked it again, and consider that it was inappropriate to put the transcriptome here. Therefore, we have deleted its relevant description. In fact, these proteins interacted with NtMMP-9 are derived from the predicted results using STRING 10.5 (http://string-db.org/). Also, the molecular docking has been re-explained in the revised version.
Results about “Heterologous expression and the enzyme activity of NtMMP-9” figureis absent, also the meaning of these results are not clearly explained. I would recommend performing an antibacterial assay.
Response: Thank you for reminding us. After the expression vector pET-28a/aNtMMP-9 was completely recombined, we transformed it into competent cells [Rosetta-gami (DE3), BL21 (DE3), and BL21 (DE3) pLysS], respectively (Figure S1 in the Supplementary materials). Briefly, only the competent cells of E. coli Rosetta-gami (DE3) grew on LB plates with kanamycin antibiotic. The positive one containing pET-28a/aNtMMP-9 vector was confirmed by PCR with the amplified fragment of 1,716 bp (Figure S2 in the Supplementary materials).
The enzyme activity of NtMMP-9 is usually measured by gelatin zymography (Tajhya et al., 2017; Cancemi et al., 2019). In this study, the manifesting band intensity reflects the protease activity, as shown in Figure S5C in the Supplementary materials.
These results are to confirm the protease activity of NtMMP-9, which is the basis to degrade extracellular matrix (ECM), alter cell-cell and cell-ECM interactions, and cleave membrane proteins of cell surface and cleave proteins in the extracellular environment. Also, these data may play an important role in expounding biological processes (Lines 43-45 in the revised version).
Thanks for your good suggestion on further performing an antibacterial assay. Based on the activities mentioned above, MMP-9 may play a role in many aspects, including cancer cell migration and invasion, inflammation, proliferation, and bacterial invasion to fish. Clearly, it is interesting and meaningful to reveal the antibacterial activities of NtMMP-9. However, our aim of this work is to confirm its proteolytic cleavage activity by degrading the gelatin. An apparent band appeared in Figure S5C, showing that NtMMP-9 has a proteolytic activity. Thus, performing an antibacterial assay is our on-going study.
References:
Cancemi P, Di Falco F, Feo S, et al. The gelatinase MMP-9like is involved in regulation of LPS inflammatory response in Ciona robusta. Fish & shellfish immunology, 2019, 86: 213-222.
Tajhya RB, Patel RS, Beeton C. Detection of matrix metalloproteinases by zymography[M]//Matrix Metalloproteases. Humana Press, New York, NY, 2017: 231-244.
Incomplete and wrongly written figure legends all throughout the text. Statistical tests used should be indicated.
Response: Thanks for your good suggestions. In the revised version, all incomplete and wrongly written figure legends have been extensively modified; and the statistical analysis has also presented (Lines 104, 208-211, 225-226, 253, 266-268, and 312-313, ).
Conclusions should be further developed.
Response: The conclusions have been rewritten (Lines 369-376 in the revised version).
Minor concerns:
Several text miss-writing errors should be corrected (i.e.,changes in letter size and several words underlined throughout the text).
Response: In the revised version, the miss-writing errors have all been corrected, including letter size, underline, and spelling errors.
Line 95:sequence species origin.
Response: The sequence species origin has been described in the second part of section Methods (Lines 194-195 in the revised version).
Line 142: explain what is RATE
Response: Thank you for reminding us. We have checked the usage of “RATE” again, and consider that the “RATE” used here is improper. Thus, we have deleted it in the revised version (Lines 155-156).
Line 173: further detail the gene sequencing results, i.e.,fragments sequence lengths, etc.
Response: According your good suggestion, we have further depicted the gene sequencing results, including fragments sequence lengths, etc. (Lines 106-109 in the revised version).
Round 2
Reviewer 1 Report
The manuscript is in a better state than the first submission. Authors have addressed all Queries raised.